# Preparation a High-Performance of Gangue-Based Geopolymer Backfill Material: Recipes Optimization Using the Taguchi Method

**DOI:** 10.3390/ma16155360

**Published:** 2023-07-30

**Authors:** Sen Yang, Hongguang Zhu, Sen Pang, Zaijie Ruan, Sinuo Lin, Yi Ding, Pengpeng Cao, Zhengyan Shen

**Affiliations:** 1School of Mechanics & Civil Engineering, China University of Mining and Technology (Beijing), Beijing 100083, China; yscumtb@163.com (S.Y.); zqt2100602052@student.cumtb.edu.cn (Z.R.); zqt2100602044@student.cumtb.edu.cn (S.L.); zqt2100620152@student.cumtb.edu.cn (Y.D.); c2819322724@163.com (P.C.); bqt2000602027@student.cumtb.edu.cn (Z.S.); 2Beijing Building Research Institute Corporation Limited of CSCEC, Beijing 100076, China; pang.sn@buaa.edu.cn

**Keywords:** gangue-based geopolymer backfill material, workability, strength behavior, Taguchi method, microstructure

## Abstract

The strip filling method in underground reservoir needs high strength to achieve the requirements of water storage. In order to address the challenges associated with costly and weak filling materials, this study aimed to develop an economically efficient and high-strength gangue-based geopolymer backfill material (GBGBM). To achieve this, the Taguchi method was employed to design a series of 25 experiments, each consisting of four factors and five levels. This study focused on investigating the effects of different gangue gradation levels, sand ratios, water binder ratios (w/b), and aggregate binder ratios (a/b) on the working characteristics and unconfined compressive strength (UCS) of the GBGBM. The optimal combination of the GBGBM was determined by employing a signal-to-noise ratio (S/N)-based extreme difference and variance analysis. The results revealed that the w/b ratio exerted the most substantial influence on both the slump and UCS. Specifically, when employing a gradation of 50%, a sand ratio of 55%, an a/b ratio of 2.5, and a w/b ratio of 0.64, the slump measured 251.2 mm, the UCS at 3d reached 5.27 MPa, and the UCS at 28d amounted to 17.65 MPa. These findings indicated a remarkable improvement in early UCS by 131.14% and the late UCS by 49.45% compared to gangue-based cement backfill material (GBCBM). Additionally, this study examined the hydration products and microstructures of both GBGBM and GBCBM using X-ray diffraction (XRD), scanning electron microscopy (SEM), and mercury intrusion porosimetry (MIP) analyses. Significantly, the GBGBM exhibited notable advantages over the GBCBM, including a 78.16% reduction in CO_2_ emissions, a 73.45% decrease in energy consumption, and a 24.82% reduction in cost. These findings underscore the potential of GBGBM as a sustainable and cost-effective alternative to GBCBM.

## 1. Introduction

The development of coal resources has played a significant role in the rapid growth of China’s national economy. However, the main coal-producing regions in China, such as Jin, Shaanxi, Meng, Ning, and Gan, are located in the western parts of the country, which have delicate ecological environments and face water scarcity challenges [1]. In light of the imperative to conserve water during coal mining operations [2], the implementation of underground water reservoir technology in coal mines offers a viable solution for efficient water resource utilization and ecological environmental protection. This approach allows for the protection of water resources while extracting coal. Studies have been conducted to utilize mine waste materials, such as coal gangue, to construct strip fillers that can partially replace coal pillars, serve as support structures, create water storage spaces, reduce the width of coal pillars, and increase the coal recovery rate [3]. However, the main purpose of backfill mining in existing coal mines is to control the breaking of key overburden rock layers and alleviate surface subsidence. The early UCS (3d) of filling materials is required to be lower than 1 MPa [4], and the later UCS (28d) is generally lower than 5 MPa [5,6,7]. Fluidity and early strength are the most important components to guarantee the stability of the dam body, the safety of coal mine production, and increase efficiency. Previous studies have attempted to meet these flowability and high strength requirements by increasing the amount of water reducer [8] and cement [9]. However, this approach proves to be prohibitively costly [3]. Furthermore, cement manufacturing results in substantial CO_2_ emissions [10] and environmental pollution [11]. Hence, it is imperative to explore eco-friendly alternatives to cement [12] and develop high-strength, flowable filling materials capable of partially replacing and enclosing coal pillar dams, by harnessing the combined advantages of coal fill mining and underground reservoirs, and by protecting and utilizing groundwater resources in ecologically fragile mining areas. This endeavor holds immense significance in realizing environmentally sustainable coal mining practices.

In recent years, there has been significant progress in the development of new mining cementing materials, with alkali excitation technology emerging as a prominent research focus in the field of mine filling. This technology offers dual advantages in terms of performance enhancement and solid waste utilization [13]. The alkali activation of waste materials (e.g., metakaolin [14,15], coal gangue, fly ash, or slag) has become an important area in the development of new green cements (inexpensive and environment friendly) and can be formulated by wastes and activators. Under extremely high OH^−^ concentrations, the Si-O-Si- and -Si-O-Al-vitreous body structure is rapidly dissolved into a solution to form the [SiO_4_]^4−^ and [AlO_4_]^5−^ tetrahedral units. Meanwhile, the new-O-Si-O-Al-O- binding materials with a three-dimensional network structure are obtained by shrinking and polymerization reaction [16]. Studies have demonstrated that geopolymers exhibit notable characteristics such as high strength, erosion resistance, impermeability, and good excellent long-term workability [17]. Both domestic and international scholars have sought to improve the flowability [18] and strength development [19] of filling materials through various measures, including the use of admixtures [13,20], adjustments to the amount of gelling material [21,22], and the selection of excitation agents [19]. However, many of these methods can be costly. In contrast, improvements from the perspective of aggregates are relatively inexpensive. Surprisingly, there is limited research exploring the influence of aggregates on the fluidity and strength of geopolymer filling materials. Qi et al. [13] studied the influence of sand ratio on the UCS and fluidity of a filling body and obtained the optimum value range of the gangue sand ratio between 50% and 70%. From the perspective of aggregates, Yu et al. [23] studied the corresponding relationship between aggregate grading, particle size proportion, and dosage on the slump and strength of the filling body. These useful conclusions suggest that the fluidity of filling materials should focus on coal gangue aggregates. Therefore, regulating the strength of filling materials from the aggregates’ perspective may offer a cost-effective approach. Consequently, the preparation of high-fluidity filling materials and the means to regulate their strength have become crucial research areas of focus. 

The Taguchi method is an experimental approach widely employed for optimizing the performance of products or processes. It offers the advantage of minimizing the impact of uncontrollable factors (such as noise or errors) on the variability of test data. By utilizing a S/N ratio, the Taguchi method aims to obtain precise test results with a minimal number of trials. Many researchers have utilized the Taguchi method in concrete design. For instance, Riahi et al. [24] applied the L9 Taguchi array to design a three-factor, three-level experiment and used analysis of variance (ANOVA) to identify the primary factors influencing the early strength of fly ash ground aggregates. Similarly, Hadi et al. [25] utilized the Taguchi method to determine the optimal mix ratio of geopolymer concrete using slag powder as the aluminosilicate source under normal temperature curing conditions. In conclusion, it has been demonstrated that it is possible to design concrete mix proportions using the Taguchi approach. Given the unique production conditions downhole, it is necessary to mix the filling slurry on the surface, ensuring excellent stability and fluidity (150–250 mm) for smooth and efficient pumping to the mining area [26]. Early strength plays a critical role in maintaining dam stability, ensuring mine safety, and optimizing operational efficiency. Therefore, the filling material must exhibit not only good flowability but also high strength. Specifically, achieving early UCS (3d) is essential to ensure the initial stability of the filling body [22] and to realize the efficient and safe production of the working face. In strip filling mining, it is necessary to achieve a relatively high strength for the filling material, with a minimum early strength requirement of 4 MPa [27]. The late UCS (28d) plays a crucial role in controlling roof subsidence and surface deformation. Many researchers have adopted the overburden bearing theory [28] and determined the strength of the backfill material based on the UCS of the surrounding rock pillar [29], with the highest UCS reaching 16 MPa [30]. In this study, optimization has two objectives: The GBGBM reaches its maximum compressive strength after 3 and 28d. Additionally, the target slump expectation value is set at 250 mm. 

The primary objective of this study is to determine the optimal combination of GBGBM by considering the effects of grading, sand ratio, w/b ratio, and a/b ratio on the slump, as well as the 3d and 28d UCS of the GBGBM. This study aims to analyze the range and variance of each factor on the slump and strength by utilizing the S/N ratio. To provide a comprehensive understanding of GBGBM, this study further investigates the microscopic hydration mechanism through the application of the XRD, MIP, and SEM techniques. By examining the ideal combination, this study aims to establish a detailed understanding of the hydration process at a microscopic level. Additionally, this study assesses the environmental and economic benefits of GBGBM by calculating carbon dioxide emissions, energy consumption, and production costs associated with cement. This evaluation contributes to a holistic analysis of GBGBM’s advantages over traditional cement-based materials. Overall, the findings of this study hold significant implications for promoting the widespread adoption of coal gangue geopolymer filling material in backfill goaf applications, thus contributing to environmental protection efforts.

## 2. Materials and Methods

### 2.1. Materials

#### 2.1.1. Aluminosilicate Materials

The following are the physical performance indicators for the test materials: The cement materials used in this study included ordinary Portland cement (P.O 42.5) provided by Quzhai Cement Co., Ltd., Shijiazhuang City, Hebei Province, China. It had a specific surface area of 345 m^2^/kg and a density of 3.15 g/cm^3^. The slag and fly ash used were sourced from Gongyi Zhongxin mineral products Co., LTD, Zhengzhou City, Henan province. The slag used had a specific surface area of 550 m^2^/kg and a density of 2.82 g/cm^3^. It was classified as S105 according to the specifications outlined in GBT18046-2008 [31]. The fly ash had a density of 2.42 g/cm^3^ and a specific surface area of 430 m^2^/kg. The calcium carbide residue (CCR) used in this study was obtained from a calcium carbide acetylene gas plant located in Hunan. The CCR had a density of 1.32 g/cm^3^ and a specific surface area of 435 m^2^/kg.

The HELOS-OASIS (New Patek, Königsbrunn, Germany) instrument was used to test the particle size distribution of different materials. As shown in Figure 1, the median particle size (d50) of the fly ash, slag, and calcium carbide residue were 15.11 μm, 7.95 μm, and 13.78 μm, respectively. XRF was used to detect the composition of aluminosilicate materials, and the results are summarized in Table 1. According to the XRD patterns in Table 1 and Figure 2, the content of SiO_2_ and Al_2_O_3_ in fly ash is high, which is mainly the crystalline phase of mullite and quartz. The slag has a high CaO content and a wide dispersion peak at 25~35°. The amorphous phase is the main component of the slag. The crystallization peaks of calcium hydroxide and calcium carbonate can be seen in the XRD pattern of the CCR. This research shows that the high alkalinity of carbide slag can be used as a solid activator and reacts with fly ash similar to volcanic ash.

#### 2.1.2. Gangue Aggregate Characteristics

The fresh gangue used in this study was obtained from a mine in Jinzhong, Shanxi Province. The XRD analysis (Figure 3) revealed that the mineral composition of the gangue consisted mainly of quartz (58.5%), kaolinite (28.5%), illite (8.49%), and pyrite (3.2%). A thin segment of the gangue sample (Figure 4) and SEM (Figure 5) provided further insights. The gangue exhibited a blocky, muddy structure with a high concentration of clay minerals, particularly clayey and argillaceous components. Silt particles were present in small amounts and dispersed throughout the gangue (Figure 4). Within the gangue, there were siliceous spongy bone needle-like cryptocrystalline silica. These needles were elliptical, rounded, and irregularly scattered. Additionally, the gangue contained a notable presence of granular pyrite and streaky organic matter residue. The gangue exhibited little apparent pore space and contained a minor amount of charcoal debris. Figure 5 also indicates the presence of clay minerals, pyrite, microporous fractures, and the banded enrichment of pyrite within the coal gangue. The internal structure of the gangue aggregate appeared loose, and weak joint surfaces such as organic matter and clay were mixed in.

The chemical composition and physical properties of the coal gangue used as coarse and fine aggregates for filling materials are provided in Table 1 and Table 2, respectively. To prepare the coal gangue for use, it was broken using a jaw crusher and then screened using a standard vibrating screen machine (ZBSX-92A type) to obtain different grain grades (0~15 mm).

#### 2.1.3. Activator

The CCR used in this study was obtained from a calcium carbide acetylene gas plant located in Hunan. The chemical composition of the slag is presented in Table 1. The liquid sodium silicate solution utilized in the experiments was produced by Bengbu Jingcheng Chemical Co. LTD. It had a modulus ranging from 3.2 to 3.5. The SiO_2_ modulus (n) of the alkali activator refers to the adjusted molar ratio of SiO_2_ to Na_2_O using a NaOH solution. For this particular study, the SiO_2_ modulus (n) was set to 1.3 [32]. Distilled water was employed for all testing purposes, and the alkali solution was prepared 24 h prior to conducting the experiments. 

### 2.2. Experimental Scheme

#### 2.2.1. Research and Development Concepts for GBGBM

The fly ash/slag mixture as the aluminosilicate precursor was activated with two kinds of activators to prepare the GBGBM’s binders, and the mass ratio of fly ash to slag was fixed at 7:3. The activator CCR and Na_2_SiO_3_ were proportioned by mass of the aluminosilicate precursor at 5 wt% and 2 wt%, respectively. Meanwhile, the GBCBM’s binder was OPC.

The preparation process of the GBGBM consisted of two main steps, as illustrated in Figure 6. The first step was to prepare the materials (gangue, fly ash, slag, and activator), design the Taguchi test scheme, test and evaluate the GBGBM’s performance, and obtain a better material ratio. In the second step, based on the optimal ratio obtained from the first step, GBCBM was prepared using cement as the cementing material. Samples were then tested for compressive strength and slump at 3d and 28d to evaluate the performance of the GBCBM. Furthermore, the chemical composition and microstructure changes of both the GBCBM and the GBGBM at 28d were analyzed using techniques such as XRD, MIP, and SEM.

#### 2.2.2. Design of Experiments by the Taguchi Method

The Taguchi method is an experimental approach aimed at optimizing the performance of a product or process. It provides a method for analyzing experimental results determining the optimal values for each measured parameter [33]. One of the advantages of the Taguchi method is its robustness against noise, which refers to (uncontrollable factors) that can affect the experimental outcomes. For optimization purposes, the Taguchi method specifies three types of parameters (noise factors) [25]:(i)Smaller the better: the goal of optimization is to minimize the reaction (e.g., reduce water secretion rate);(ii)Larger the better: the objective is to maximize the response (e.g., maximize the UCS);(iii)Nominal is best: the goal is to reach a target value, such as the slump target value.

To interpret the experimental results, the S/N ratio is calculated for each experiment. The “signal” represents the desired response (e.g., UCS), while the “noise” corresponds to the set of parameters for a particular experiment. The specific definition of the S/N ratio depends on the optimization objective.

In this study, the S/N ratio was used to evaluate the UCS and slump characteristics of GBGBM. Equations (1) and (2) were employed to calculate the S/N ratios for the respective characteristics.
(1)$$S/N=-10\log_{10}\left[\frac{1}{n}\sum_{i=1}^{n}\frac{1}{y_{i}^{2}}\right]$$
(2)$$S/N=10\log_{10}(\frac{\frac{1}{n}\sum_{i=1}^{n}y_{i}}{\frac{1}{n-1}\sum_{i=1}^{n}{\left(y_{i}-\bar{y}\right)}^{2}})$$
where y_i_ is the test observation (UCS and slump), n depicts the number of test repetitions, S/N is the signal-to-noise ratio, and $\bar{y}$ represents the mean of the experimental data (slump). The Taguchi design, variance analysis, main effect analysis, and the calculation of relevant statistical indexes were carried out by the Minitab 15.0 software.

Four main parameters, including gradation (100%, 75%, 50%, 25%, and 0%), sand ratio (40%, 45%, 50%, 55%, and 60%), w/b (0.6, 0.62, 0.64, 0.66, and 0.68), and a/b (2, 2.25, 2.5, 2.75, and 3) were considered in the mix design (Table 3). A total of 25 trial mixes were prepared depending on the L25 array obtained using the Taguchi method, in accordance with Table 4.

#### 2.2.3. Specimen Preparation and Testing

The GBGBM mixing process followed the Taguchi test procedure. Each sample was initially mixed in a concrete mixer for 60 s. Then, an alkali solution was added to the mixture. To achieve the desired fluidity, the samples were further mixed for an additional five minutes. A slump test was performed on a portion of the slurry to assess its workability. The remaining portion of the slurry was cast into cubic plastic molds measuring 100 × 100 × 100 mm. After 24 h, the specimens were removed from the molds and placed in a temperature-constant humidity curing box (20 °C, RH = 90%). The specimens were then either cured for 3d or 28d. An MTS GTC350 electro-hydraulic servo-controlled testing system, following the same loading rate (1 mm/min) as conducted by Zhou [34], was used to test the UCS of the specimens. After curing for 28d, the dried specimens were cut into small rectangular specimens (height = 5–10 mm, length = 10 mm, width = 10 mm) with the naturally formed surface. Micrographs of the prepared scanning electron microscope (SEM) specimens were captured using a Quanta FEG450 scanning electron microscope, a BRUCKER D8-ADVANCE X-ray diffractometer (XRD), and an AutoPore V9600 Micrometrics Instrument (MIP) to examine the hydration products and microstructures [32]. 

## 3. Results and Discussion

Equations (1) and (2), along with the test results presented in Table 3, were utilized to calculate the mean S/N response values for the GBGBM’s slump and UCS, as displayed in Table 4. Additionally, microscopic testing techniques such as XRD, MIP, and SEM were employed to investigate the microscopic hydration process of the GBGBM under the ideal combination.

### 3.1. Range Analysis

#### 3.1.1. Range Analysis of Slump S/N of GBGBM

The orthogonal design employed in this study allowed for the assessment of the degree of influence of each element, with a larger extreme difference indicating a more significant impact of the factor. By analyzing the average S/N ratio and range R of the four factors gradation, sand ratio, w/b, and a/b at each level, it was observed that the w/b ratio exerted the greatest influence on the slump of the GBGBM, followed by the a/b ratio, while the impact of gradation was relatively minor. Therefore, the order of significance of the four factors affecting the GBGBM’s slump is as follows: w/b ratio > a/b ratio > sand ratio > gradation. This finding further emphasizes the dominant role of the w/b ratio in determining the slump of the GBGBM.

According to the S/N ratio data in Table 4, the range of the slump and the UCS’s S/N ratio was obtained. Based on the range analysis results presented in Table 5, the variation trend chart for each factor at different levels was generated, as depicted in Figure 7a. When the gradation was of 50%, the sand ratio of 55%, the w/b ratio of 0.68, and the a/b ratio of 2, the S/N ratio of the slump reaches the maximum. Therefore, the optimal combination determined using the Taguchi test method is as follows: a gradation of 50%, a sand ratio of 55%, a w/b ratio of 0.68, and an a/b ratio of 2. After conducting supplementary verification tests, the S/N ratio of the slump for the optimal combination was found to be 56.59 dB, with an average slump value of 276.5 mm.

In Figure 7a, it can be observed that as the w/b ratio increased, the slump’s S/N ratio also increased. However, it should be noted that if the w/b ratio exceeded 0.66, issues such as segregation and water secretion might have occurred. On the other hand, as the a/b ratio increased, the slump’s S/N ratio decreased. This was because a higher a/b ratio reduced the amount of cementitious material and slurry surrounding the aggregate, leading to poor compatibility. Regarding the effect of gradation and the sand ratio, the slump’s S/N ratio tended to initially increase and then decrease. This can be attributed to the competing processes of the growth of the encapsulation thickness and the decrease in specific surface area of the aggregate. When the sand ratio increased, the void ratio decreased, and the total surface area of the aggregate decreased. This resulted in the formation of a thicker encapsulation layer and improved flow. However, when the sand ratio exceeded 55%, more slurry was needed to fill and wrap the aggregate, which reduced the amount of slurry available for lubrication and decreased the fluidity of the GBGBM mix. Similarly, changes in gradation affected the voids. When the gradation was 50%, the voids were the smallest. A reasonable gradation reduced the resistance during the pipeline transport of the slurry, decreased the amount of cementing material required, and promoted the utilization of gangue.

#### 3.1.2. Range Analysis of UCS’s S/N Ratio of GBGBM

As evident from the UCS’s S/N ratio range in Table 5, the factors affecting the 3d UCS of the GBGBM in the test showed the following range R in descending order: w/b ratio > a/b ratio > gradation > sand ratio. Similarly, the range analysis of the 28d UCS revealed the following order: w/b ratio > a/b ratio > sand ratio > gradation.

The main effect plots for the S/N ratio of the 3d and 28d UCS are shown in Figure 7b,c, respectively. The w/b ratio exhibited the largest difference in the UCS’s S/N ratio study, indicating that it was the primary factor influencing both the 3d and 28d UCS. The greater the range is, the stronger the impact of this factor, and therefore, the more important it is [35].

Figure 7b shows that among the 25 sets of GBGBM specimens, the highest S/N ratio for the 3d UCS was achieved by GBGBM-1, which had a grading of 100%, a sand ratio of 40%, a w/b ratio of 0.6, and an a/b ratio of 2. Based on the Taguchi test method, the 3d optimal combination for GBGBM was a grading of 100%, a sand ratio of 55%, a w/b ratio of 0.6, and an a/b ratio of 2. Therefore, the 3d S/N ratio was 16.76 dB, and the UCS was 5.27 MPa. Following the additional verification test, the slump measurement resulted in 25.49 dB, corresponding to a mean value of 18.52 MPa for the 28d UCS. Additionally, the gradation S/N ratio for the 28d UCS was 50%, while the sand ratio was 55%, the w/b ratio was 0.6, and the a/b ratio was 2.5.

At 3d and 28d, the UCS displayed a decline when the w/b ratio escalated, as illustrated in Figure 7b,c. The 28d UCS exhibited an initial increase followed by a subsequent decrease as the a/b ratio expanded, achieving its maximum value at an a/b ratio of 2.5. Conversely, the 3d UCS decreased with an increase in the a/b ratio. The strength S/N ratio experienced an ascent with a arising sand ratio, reaching its peak at a sand ratio of 55%, which coincided with the critical point of the UCS for both the 3d and 28d S/N ratio. While the 28d UCS exhibited an inflection point at a grade of 0.5, when the UCS’s S/N ratio was at its greatest and the strength change turned, the 3d UCS declined as the grade increased. The analysis of extreme differences presented in Table 4 provided additional evidence to substantiate the observation that the sand ratio at 28 d exhibited superior performance compared to the early UCS’s S/N ratio based on gradation. As the grading level decreased, there was an increase in the ratio of the maximum particle size of the coarse aggregate, resulting in a decrease in the specific surface area of the coarse aggregate. This decrease in the specific surface area led to a reduction in the amount of early hydration products. Consequently, there was a decrease in the interfacial bonding force and an increase in the interfacial transition zone, which represents the weakest point in the concrete structure. This phenomenon represented the weakest aspect in the concrete structure, hence necessitating a decrease in the maximum particle size ratio of the coarse aggregate. In the later stage, as the cementing material underwent complete hydration, the cementing effect was enhanced, and a well-balanced aggregate gradation substantially mitigated the occurrence of surface irregularities such as the honeycomb and hemp surface phenomenon. Consequently, the particle gradation of the aggregate exerted a significant influence on the ultimate UCS of the filling material.

### 3.2. Variance Analysis

In order to further evaluate the impact of various factors on the slump and the UCS of GBGBM, a variance analysis was carried out on the test data, and the contribution of each factor was obtained accordingly. Table 6 presents the significance and the relative contribution of each factor to the variance in the GBGBM’s compressive strength and slump based on the mean variance.

Table 6 provides evidence of the significant impacts of the three variables on the decline of the GBGBM, as indicated by the *p*-values below 0.05 or a significance at the 95% confidence interval. Specifically, the w/b and a/b ratios exhibited a highly significant influence on the slump of the GBGBM were highly significant, with a *p*-value of 0.001, which was lower than 0.01, signifying significance at the 99% confidence level. Moreover, in the 28d UCS, the effects of grading, the sand ratio, the w/b ratio, and the a/b ratio were all highly significant with a *p* -value of 0.001, which was below 0.01, indicating significance at the 99% confidence interval. Similarly, all other factors exhibited highly significant effects on the 3d UCS, with the exception of the sand ratio, which was still significant. This analysis demonstrates that, except for the gradation, the changes in the levels of each element had effects on the GBGBM’s slump and strength S/N ratio that were within the margin of error. The contributions from grading, the sand ratio, the w/b ratio, and the a/b ratio to the slump were 8.29%, 16.30%, 30.89%, and 21.01%, respectively. However, the level of influence varied for each major element. The ratios of grading, the sand ratio, the w/b ratio, and the a/b ratio contributing to the 3d UCS were 12.48%, 3.16%, 66.29%, and 13.51%, respectively. For the 28d UCS, the contributing grading ratio, the sand ratio, the w/b ratio, and the a/b ratio were 10.87%, 13.94%, 49.25%, and 16.23%, respectively. The results of the extreme difference analysis, which all showed that the w/b ratio is the factor that most significantly affects the slump and the UCS of GBGBM, are in line with the degree of their influence. 

### 3.3. Optimal Combination

Among all the factors considered, the w/b ratio made the largest contribution of 30.89% towards the slump’s S/N ratio in the GBGBM. This indicates that it is the most significant component influencing the slump, as supported by the extreme difference analysis, factor index analysis, and variance analysis. Although the highest slump achieved by the ideal slump combination based on the slump’s S/N ratio was 276.5 mm, the flowability declined, and aggregate sinking and segregation took place (Figure 8c). To examine the impact of different w/b ratios on the slump, the experiment kept the gradation, a/b, and sand ratios constant while varying the w/b ratio. When the w/b ratio was set at 0.6, the slump’s S/N ratio was measured at 45.51 dB, and the corresponding slump value was 234.7 mm (Figure 8a). When the w/b ratio was set at 0.62, the S/N ratio and slump were 49.46 dB and 242.7 mm, respectively. These values were close to the desired value of 250 mm. At a w/b ratio of 0.64, the S/N ratio was measured at 49.72 dB, and the corresponding slump was 256.9 mm, demonstrating good compatibility and cohesion qualities (Figure 8b). When the w/b ratio was 0.66, the S/N ratio and the slump were 50.96 dB and 273.6 mm, respectively, showing the same rule as in Figure 8c. The optimal slump combination based on the w/b ratio was as follows: a grading of 50%, a sand ratio of 55%, a w/b ratio of 0.64, and an a/b ratio of 2. After conducting supplementary verification tests, the S/N ratio of the slump for the combination was found to be 49.72 dB, with an average slump value of 256.9 mm, which met the fluidity requirements of the filling body.

Based on the range analysis, the factor index analysis, and the variance analysis of the intensity of the S/N ratio at 3d and 28d, it was determined that the w/b ratio remained the most influential factor in determining the UCS’s S/N ratio, and the UCS contribution rate at 3d and 28d was 66.29% and 49.25%, respectively. Using the same method as the slump, the w/b ratio was changed to achieve the optimal combination of the relevant UCS. In the analysis of the UCS’s S/N ratio at 3d and 28d, the sand ratio and the w/b ratio were the same. In the UCS’s S/N ratio at 3d, the a/b ratio was 2 while that at 28d was 2.5. To reduce costs and improve the utilization rate of coal gangue, the a/b ratio was set at 2.5. By considering the slump’s S/N ratio as well as the 3d and 28d UCS’s S/N ratio, the optimal UCS combination of the GBGBM was determined as follows: a grading of 50%, a sand ratio of 55%, a w/b ratio of 0.64, and an a/b ratio of 2.5. After conducting supplementary verification tests, the slump’s S/N ratio was found to be 47.74 dB, with a mean slump value of the slump at 251.2 mm. The 3d UCS’s S/N ratio was 14. 51 dB, with a mean UCS value of 5.27 MPa. At 28d, the S/N ratio for the UCS was 25.07 dB, and the mean UCS value was 17.65 MPa. The test results of the GBCBM filling material prepared with the optimal combination ratio of the GBGBM met the requirements for fluidity, early strength, and late strength. According to the optimal combination ratio of GBGBM, the GBCBM filling material was prepared with cement paste. The measured slump was 242.5 mm, the 3d UCS was 2.28 MPa, and the 28d UCS was 11.81 MPa.

### 3.4. Analysis of Microscopic Results

#### 3.4.1. XRD Analysis

The XRD patterns of the GBCBM and GBGBM after 28d of curing are presented in Figure 9a,b, respectively. The XRD pattern was analyzed with the Jade software, and it can be seen from Figure 9a that the main characteristic peaks of the GBCBM were SiO_2_, ettringite, Ca(OH)_2_, and a C-S-H gel [36]. The main characteristic peaks of the GBGBM were SiO_2_ and Al_6_Si_2_O_13_ (near 26°) in unreacted fly ash particles, CaMg(CO_3_)_2_ from slag and calcium hydroxide (18°, 34°), and calcium carbonate (29°) in solid activators [37]. The GBGBM hydration products mainly included N-A-S-H (NaAlSi_2_O_6_•H_2_O), (about 33°) and C-(A)-S-H (CaAl_2_Si_2_O_8_•4H_2_O), (29~32°) gels [38], which were the main source of strength.

The XRD analysis in Figure 9 reveals that GBGBM exhibits significantly increased dispersion diffraction peaks in the range of 20–30° compared to GBCBM. This observation can be attributed to the formation of C-(A)-S-H and N-A-S-H gels through alkali activation of slag-fly ash [38]. The broadening of the diffraction peaks indicates the presence of a larger number of hydration products [37]. And there was a small amount of Ca(OH)_2_ in the GBGBM, which comes from CCR, while CaCO_3_ was partly from the CCR and partly from carbonization [39]. In the GBCBM, the main hydration products of cement were mainly AFt, Ca(OH)_2_ and the C-S-H gel. However, in the GBGBM, the presence of alkali stimulates the formation of the C-(A)-S-H gel and N-A-S-H gel in the slag-fly ash. These additional hydration products intertwine with each other, creating a three-dimensional network structure with a dense configuration.

The calculation results of the relative integrated intensity of the XRD are shown in Table 7. It should be noted that the relative integral strength calculated for the same phase between different samples is proportional to the content but cannot represent the actual percentage of each phase. As can be seen from Table 7, the integral intensity of the primary characteristic peaks of the GBCBM were C-S-H (1059.28), C-A-S-H (104.93), Ca(OH)_2_ (1213.29), CaCO_3_ (919.24), and AFt (246.57), indicating that the main hydration products of the GBCBM were the C-S-H gel, Ca(OH)_2_, and CaCO3. The integral intensity of the GBGBM were C-S-H (840.58), C-A-S-H (1191.27), N-A-S-H (734.28), Ca(OH)_2_ (30.96), and CaCO3 (361.74). Compared with the GBGBM, the integral intensity of Ca(OH)_2_ and CaCO_3_ in the GBCBM increased by 1182.33 and 557.50, respectively. The results show that a large amount of Ca(OH)_2_ and CaCO_3_ existed in the hydration products, resulting in a loose structure and an increased porosity of the hydration products at the interface of the mortar and aggregate. In the GBGBM, the addition of CCR improves the Ca/Si and OH^−^ concentration, accelerates the depolymerization ability of aluminosilicate precursors, and forms more C-A-S-H and N-A-S-H gels [36], especially the formation of the N-A-S-H gel with a three-dimensional network structure, which fills the gap between the C-S-H and C-A-S-H gels, making hydration products dense, and the structure at the interface became denser. As a result, the GBGBM exhibited a lower porosity and a higher UCS compared to the GBCBM. The previous section of this study confirmed that the UCS of GBGBM is 131.14% higher than that of GBCBM at 3d and 49.45% higher at 28d.

#### 3.4.2. Pore Structure Analysis of GBCBM and GBGBM

Figure 10 displays the pore size distribution (PSD) curves of GBCBM and GBGBM. The aperture corresponding to the peak value of the differential curve was the maximum aperture [40]. Figure 10 illustrates that the GBCBM exhibits a notable shift towards larger apertures, ranging from 62.47 nm to 120.87 nm, when compared to the GBGBM. This shift suggests that the cumulative aperture distribution curve has sequentially migrated towards larger apertures. To investigate the pattern of the PSD evolution, the Fan pore classification method was employed [32], and the pore size was defined as micropore, transition pore, mesopore, and macropore, corresponding to pore sizes < 10 nm, 10–100 nm, 100–1000 nm, and >1000 nm, respectively. The classification results for the total porosity and pore size are shown in Table 8 and Figure 11. In the GBCBM, the total porosity was measured as 37.13% with a median pore size of 70.67 nm, while in the GBGBM, the total porosity was recorded as 34.92% with a median pore size of 47.36 nm. Comparatively, the volume porosity of the GBGBM was found to be 5.95% lower than that of the GBCBM.

Figure 11 presents the pore volume histogram and the distribution of the different pore types. When compared to the GBGBM, the GBCBM exhibited an increase in the volume of micropores, transitional pores, and macropores by 42.10%, 26.54%, and 88.99%, respectively, indicating that the macropore volume was close to that of the GBGBM. Studies show that when the pore size is larger than 100 nm, it is defined as a harmful pore [32]. In comparison to the GBCBM, the volume of harmful pores in the GBGBM decreased by 76%, with the larger voids transforming into smaller pores. This transformation resulted in a reduction in the pore space within the matrix, improved the filling capacity of the hydration products, decreased porosity, and ultimately enhanced the strength of the backfill. This conclusion is further supported by the SEM diagram provided below.

#### 3.4.3. SEM Analysis

The microstructure of the GBCBM and GBGBM at 28d are shown in Figure 12. As can be seen from Figure 12a, the GBCBM sample was mainly characterized by the amorphous C-S-H gel, acicular ettringite, and micro-cracks. The concentration of ettringite was found to be notably high near the transition zone of the aggregate and mortar interface. This phenomenon led to the formation of micro-cracks within that specific region, resulting in a less dense microstructure. As a consequence, the porosity increased, the bond at the interface weakened, and the number of transition zones at the interface multiplied. The reasons for the formation of interfacial transition zones accompanied by micro-cracks can be verified in the studies of Fan [41] and Zhu [16]. The transition zone of the interface was the weakest link in the concrete, and the strength decreased. It is worth noting that, compared with the GBCBM, the microstructure of the GBGBM was denser, the alkali excitation slag-fly ash generated more C-(A)-S-H and N-A-S-H gels [37] (Figure 12b), there was no interfacial transition zone of the GBCBM, and the C-A-S-H and N-A-S-H gel cover around the bonding interface, and the adhesion between aggregate and mortar was enhanced, with fewer interface micro-cracks, a smaller pore distribution and a lower porosity, which were the main reasons for the strength improvement compared with the GBCBM. 

### 3.5. Environmental Benefits

The comparative analysis of the GBCBM and GBGBM binders was conducted based on existing research on the environmental impact of OPC and its substitute, geopolymers [10,42,43,44,45,46,47]. This analysis focused on the production cost, CO_2_ emissions, and energy consumption associated with the two binders (Table 9). The relevant data regarding raw materials and their ratios can be found in Table 10. It is important to note that since the CCR powder was dried using a traditional process [48], the CO_2_ emissions and energy consumption associated with the CCR were not considered in this study [46]. To estimate the CO_2_ emissions and energy consumption, data from previous studies on other raw materials such as OPC, slag, fly ash, and activators were referenced [42,44,46]. The cost data used in the analysis were obtained from the Chinese market. However, it is important to mention that the transportation costs of the raw materials were not calculated in this study to simplify the calculation process.

Based on the findings presented in Table 9 and Table 10, the production cost of 1 ton of the GBCBM binder was CNY 500. Additionally, the CO_2_ emissions associated with its production were measured at 900 kg, and the energy consumption amounted to 5.5 GJ. For the GBGBM, the alkali activator constituted the primary cost component of the binder. The cost of sodium hydroxide and sodium silicate was found to be 6 times and 2.6 times that of OPC, respectively. However, it is worth noting that the CO_2_ emissions and energy consumption of the activator were lower compared to those of OPC. The production cost of 1 ton of the GBGBM binder was determined to be CNY 377.59, with CO_2_ emissions amounting to 196.56 kg and the energy consumption totaling 1.46 GJ. Despite the higher cost of the alkali exciter compared to OPC, the use of an alkali exciter in binder preparation resulted in a cost reduction of 24.82% for the overall binder production. Additionally, there was a significant reduction in CO_2_ emissions by 78.16% and energy consumption by 73.45% compared to conventional OPC production. These findings highlight the potential economic and environmental benefits associated with the use of alkali exciters in binder formulations. The alkali-activated material demonstrated superior early and late strength compared to OPC, with increases of 131.14% and 49.45%, higher than the GBCBM, respectively. In conclusion, the GBGBM exhibited favorable characteristics such as a low carbon footprint, low energy consumption, and cost effectiveness. The successful combination of an activator and CCR showcased its applicability as a mine filling material, offering excellent economic benefits and a high bond strength.

## 4. Conclusions

The optimization of the process parameters for GBGBM was performed via Taguchi’s parameter design method. An L25 orthogonal array was used to accommodate four control factors and each with five levels for the experimental plan. The selected process parameters along with their levels were as follows: gradation (100%, 75%, 50%, 25%, and 0%), w/b ratio (0.6, 0.62, 0.64, 0.66, and 0.68); sand ratio (40%, 45%, 50%, 55%, and 60%); and a/b ratio (2, 2.25, 2.5, 2.75, and 3). The main conclusions are as follows: (1)Taguchi’s approach yielded the following ideal ratios as the best combination: a gradation of 50%, a sand ratio of 55%, a w/b ratio of 0.64, and an a/b ratio of 2.5. As a consequence, the GBGBM’s slump was 251.2 mm, and the UCS at 3d and 28d was 5.27 MPa and 17.65 MPa, respectively. These findings indicate a remarkable improvement in early UCS by 131.14% and the late UCS by 49.45% compared to the GBCBM.(2)The results of the contributions analysis all showed that the w/b and a/b ratios were the factors that most significantly affect the slump and the UCS of the GBGBM.(3)The reason for the higher UCS of the GBGBM compared to the GBCBM can be attributed to the formation of more C-(A)-S-H and N-A-S-H gels. This is further supported by the lower porosity and denser structure observed in the GBGBM compared to the GBCBM.(4)The production of the GBGBM has demonstrated a significant reduction in CO_2_ emissions, energy consumption, and costs when compared to the conventional GBCBM.

Finally, this paper prepared a high-performance coal gangue geopolymer filling material with high fluidity, excellent economic benefits, and a high bond strength. It provides a reference for long-term applications in complex mine water environments. 

## Figures and Tables

**Figure 1 materials-16-05360-f001:**
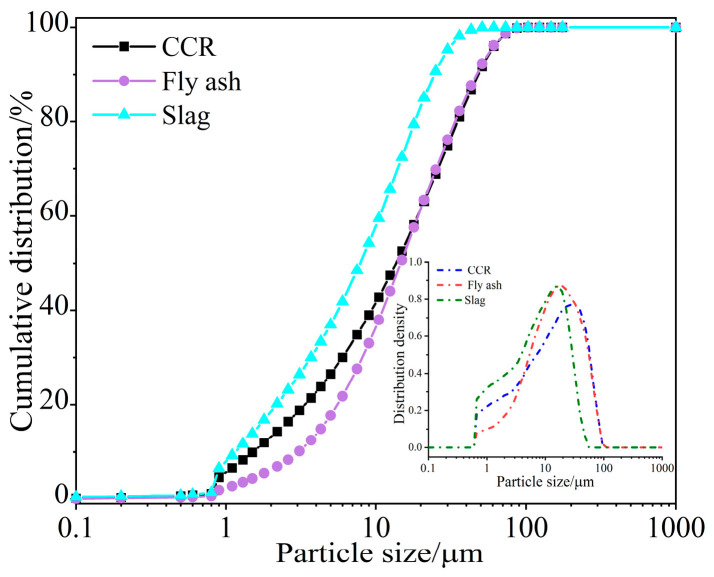
Grain size distribution of CCR, slag, and fly ash.

**Figure 2 materials-16-05360-f002:**
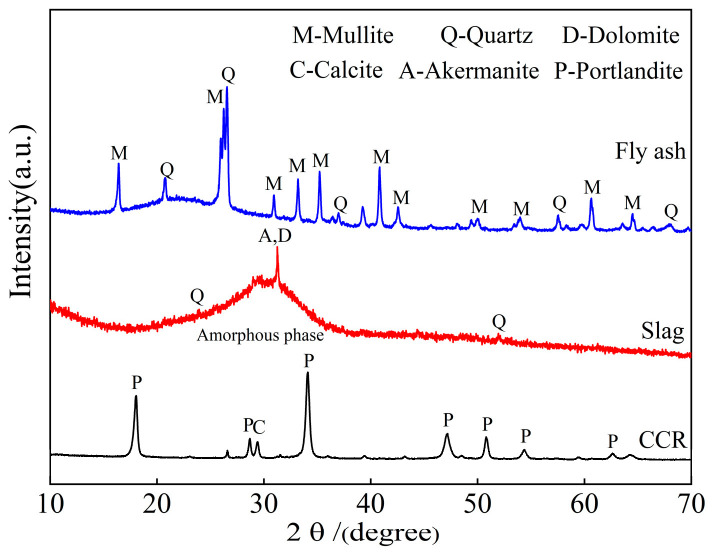
XRD of CCR, slag, and fly ash.

**Figure 3 materials-16-05360-f003:**
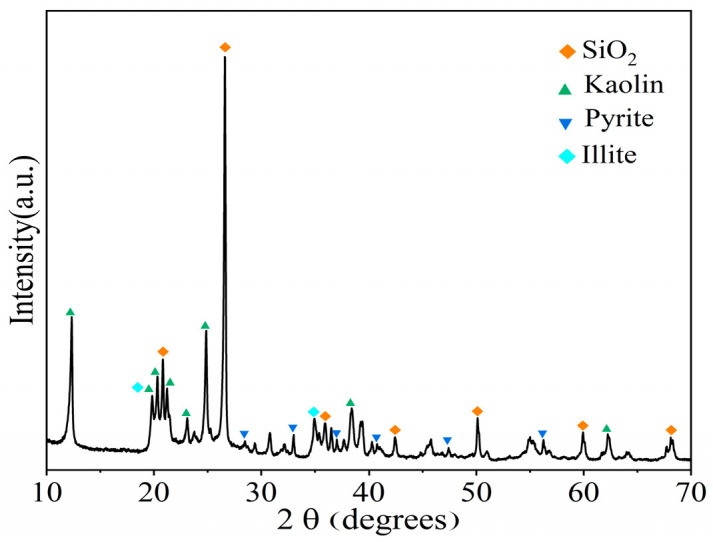
XRD of gangue.

**Figure 4 materials-16-05360-f004:**
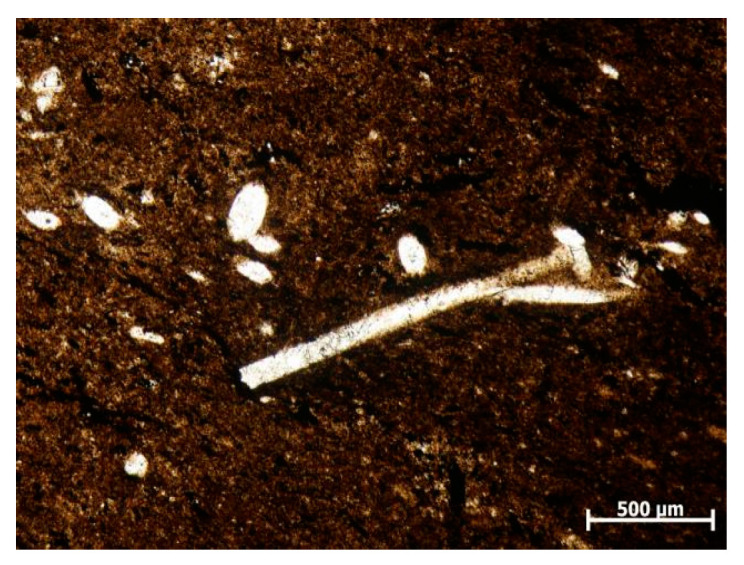
Gangue cast; thin sheet image.

**Figure 5 materials-16-05360-f005:**
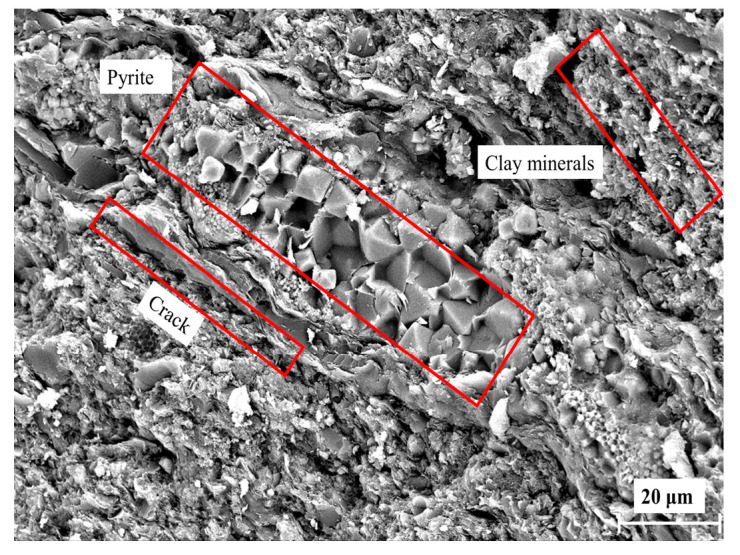
SEM of gangue (clay minerals, micro-cracks and pyrite are identified by red boxes).

**Figure 6 materials-16-05360-f006:**
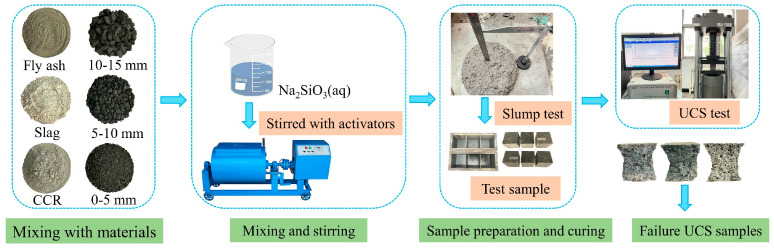
UCS and slump testing processes.

**Figure 7 materials-16-05360-f007:**
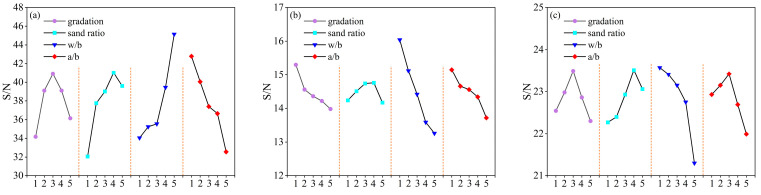
S/N ratio main effects plot for (**a**) (slump) and (**b**) 3d and (**c**) 28d UCS.

**Figure 8 materials-16-05360-f008:**
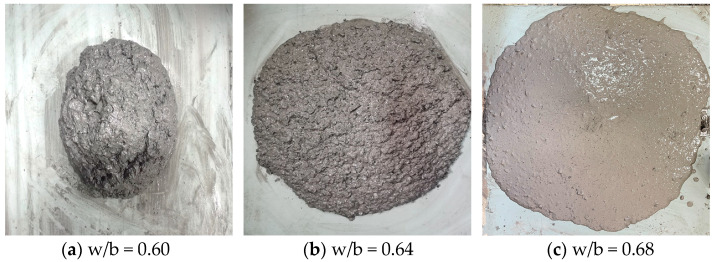
Slump under different w/b ratios.

**Figure 9 materials-16-05360-f009:**
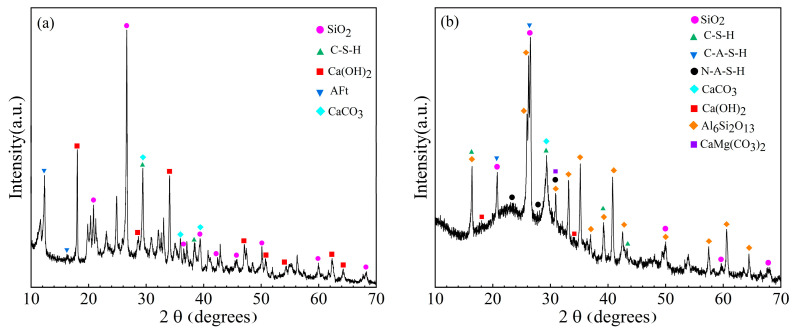
XRD patterns of (**a**) and (**b**) represent GBCBM and GBGBM, respectively.

**Figure 10 materials-16-05360-f010:**
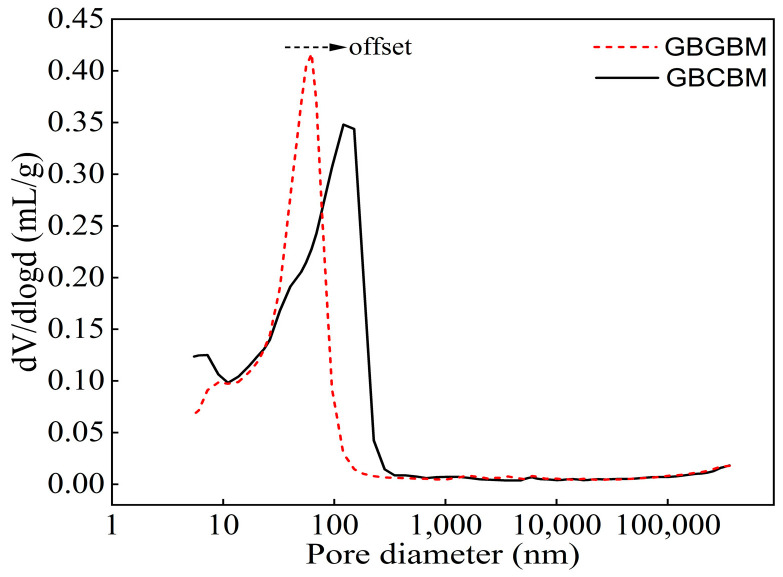
Differential PSD of the specimens.

**Figure 11 materials-16-05360-f011:**
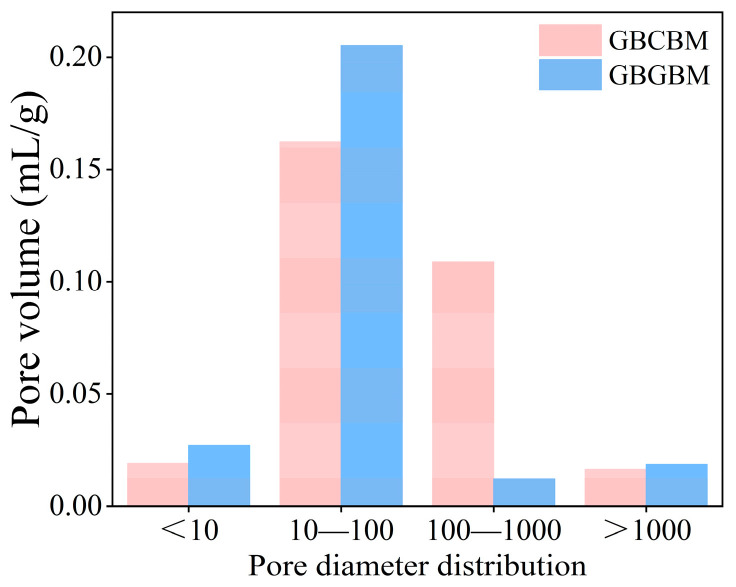
The pore volume with different pore sizes.

**Figure 12 materials-16-05360-f012:**
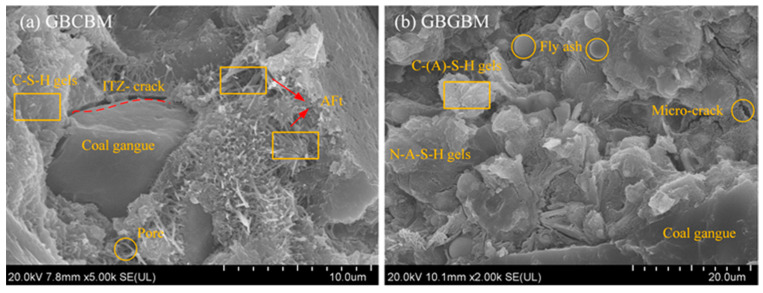
SEM images of GBCBM and GBGBM(Yellow boxes are different hydration products and unreacted particles).

**Table 1 materials-16-05360-t001:** Chemical compositions of materials (% by total mass). LOI is loss on ignition at 1000 °C.

Materials	CaO	SiO_2_	Al_2_O_3_	Fe_2_O_3_	MgO	SO_3_	K_2_O	TiO_2_	LOI
Fly ash	2.66	55.71	32.79	4.43	0.24	0.65	1.54	1.66	1.51
Slag	42.73	32.93	14.47	0.24	6.59	0.99	0.48	0.64	1.17
CCR	68.97	1.57	1.98	0.47	-	0.32	-	-	25.89
OPC	65.52	19.38	4.66	-	1.68	2.03	-	-	1.08
Gangue	0.15	70.20	17.63	1.50	0.29	-	0.88	0.45	8.51

**Table 2 materials-16-05360-t002:** Characteristics of the gangue.

Types	Apparent Density kg/m^3^	Water Content /%	Water Absorption /%	Crushing Index Value /%
Gangue	2600	0.89	4.08	13.2

**Table 3 materials-16-05360-t003:** Control factors and their levels.

Factors	Level 1	Level 2	Level 3	Level 4	Level 5
gradtion	100	75	50	25	0
sand ratio	40	45	50	55	60
w/b	0.6	0.62	0.64	0.66	0.68
a/b	2	2.25	2.5	2.75	3

**Table 4 materials-16-05360-t004:** Mix design and test results.

Specimes	Gradation	Sand Ratio	w/b	a/b	3d/MPa	28d/MPa	Slump/mm	S/N ^a^	S/N ^b^	S/N ^c^
GBGBM-1	100	40	0.6	2	7.32	14.26	244.0	32.70	17.29	23.08
GBGBM-2	100	45	0.62	2.25	6.25	13.74	238.5	33.66	15.92	22.76
GBGBM-3	100	50	0.64	2.5	5.98	14.38	246.0	35.27	15.53	23.16
GBGBM-4	100	55	0.66	2.75	5.31	14.18	236.5	40.94	14.50	23.03
GBGBM-5	100	60	0.68	3	4.62	10.83	218.1	38.19	13.28	20.69
GBGBM-6	75	40	0.62	2.5	5.77	15.14	236.0	28.88	15.22	23.60
GBGBM-7	75	45	0.64	2.75	5.31	14.29	237.0	34.94	14.51	23.10
GBGBM-8	75	50	0.66	3	4.41	12.71	232.8	39.47	12.86	22.08
GBGBM-9	75	55	0.68	2	5.18	12.54	270.5	51.65	14.28	21.97
GBGBM-10	75	60	0.6	2.25	6.27	16.18	226.5	40.57	15.94	24.17
GBGBM-11	50	40	0.64	3	4.63	13.28	208.0	25.29	13.29	22.45
GBGBM-12	50	45	0.66	2	5.22	14.02	284.0	46.06	14.36	22.93
GBGBM-13	50	50	0.68	2.25	4.79	13.29	261.5	51.36	13.61	22.47
GBGBM-14	50	55	0.6	2.5	6.39	18.92	221.4	40.97	16.11	25.54
GBGBM-15	50	60	0.62	2.75	5.29	15.97	218.6	40.86	14.46	24.06
GBGBM-16	25	40	0.66	2.25	4.64	13.42	262.5	37.41	13.33	22.55
GBGBM-17	25	45	0.68	2.5	4.48	12.12	281.3	48.47	13.02	21.67
GBGBM-18	25	50	0.6	2.75	6.40	15.07	234.0	30.39	16.12	23.56
GBGBM-19	25	55	0.62	3	5.26	14.50	218.0	34.22	14.38	23.23
GBGBM-20	25	60	0.64	2	5.16	14.57	251.0	44.98	14.25	23.27
GBGBM-21	0	40	0.68	2.75	4.03	9.66	246.8	36.05	12.10	19.69
GBGBM-22	0	45	0.6	3	5.48	11.91	162.0	25.62	14.77	21.52
GBGBM-23	0	50	0.62	2	6.03	14.79	240.0	38.57	15.59	23.40
GBGBM-24	0	55	0.64	2.25	5.32	15.46	257.5	37.25	14.51	23.78
GBGBM-25	0	60	0.66	2.5	4.42	14.34	264.0	33.38	12.91	23.13

Note: S/N ^a^, S/N ^b^, and S/N ^c^ stand for slump and 3d UCS’s and 28d UCS’s S/N ratio, respectively.

**Table 5 materials-16-05360-t005:** Range analysis of UCS and slump’s S/N ratio of GBGBM.

Response Variable	Level	Factor			
		Gradation	Sand Ratio	w/b	a/b
Slump/mm	1	34.17	32.07	34.05	42.79
	2	39.10	37.75	35.24	40.05
	3	40.91	39.01	35.55	37.39
	4	39.10	41.01	39.45	36.64
	5	36.15	39.60	45.14	32.56
	Range	6.73	8.94	11.09	10.24
3d UCS/MPa	1	15.30	14.24	16.04	15.15
	2	14.56	14.51	15.12	14.66
	3	14.36	14.74	14.42	14.56
	4	14.22	14.76	13.59	14.34
	5	13.98	14.17	13.26	13.72
	Range	1.32	0.59	2.79	1.44
28d UCS/MPa	1	22.30	22.27	23.57	22.93
	2	22.86	22.4	23.41	23.15
	3	23.49	22.93	23.15	23.42
	4	22.98	23.51	22.75	22.69
	5	22.54	23.06	21.30	21.99
	Range	1.19	1.24	2.28	1.43

**Table 6 materials-16-05360-t006:** Variance analysis of slump and 3d and 28d UCS’s S/N ratio of GBGBM.

Response Variable	Control Factor	Freedom	Sum of Square	Mean Square Sum	F Value	*p* Value	Contribution Ratio/%
	gradation	4	144.35	36.09	3.11	0.080	8.29
	sand ratio	4	239.08	59.77	5.16	0.024	16.30
Slump/mm	w/b	4	411.68	102.92	8.88	0.005	30.89
	a/b	4	294.84	73.71	6.36	0.003	21.01
	Error	8	92.72	11.59			
	Total	24	1182.68				
	gradation	4	5.08	1.27	17.45	0.001	12.48
	sand ratio	4	1.50	0.38	5.17	0.024	3.16
3d UCS/MPa	w/b	4	25.71	6.43	88.36	0.001	66.29
	a/b	4	5.47	1.37	18.81	0.001	13.51
	Error	8	0.58	0.07			
	Total	24	38.34				
	gradation	4	4.12	1.03	7.71	0.008	10.87
	sand ratio	4	5.13	1.28	9.60	0.004	13.94
28d UCS/MPa	w/b	4	16.77	4.19	31.41	0.001	49.25
	a/b	4	5.88	1.47	11.02	0.002	16.23
	Error	8	1.07	0.13			
	Total	24	32.97				

**Table 7 materials-16-05360-t007:** XRD integral intensity of different specimens.

Specimens	Mullite	SiO_2_	C-S-H	C-A-S-H	N-A-S-H	CaCO_3_	Ca(OH)_2_	AFt	CaMg(CO_3_)_2_
GBCBM	-	549.15	1059.28	104.93	-	919.24	1213.29	246.57	-
GBGBM	2382.15	779.16	840.58	1191.27	734.28	361.74	30.96	-	129.65

**Table 8 materials-16-05360-t008:** Pore volume of GBCBM and GBGBM at 28d.

Samples	Total Porosity (%)	Median Pore Diameter (nm)	Pore Volume (mL/g)			
			<10	10–100	100–1000	>1000
GBCBM	37.13	70.67 nm	0.019	0.162	0.109	0.016
GBGBM	34.92	47.36 nm	0.027	0.205	0.012	0.018

**Table 9 materials-16-05360-t009:** Production cost, CO_2_ emission, and energy consumption.

Raw Materials	Cost (CNY/t)	CO_2_ Emission (kg/t)	Energy Consumption (GJ/t)
OPC	500	900	5.5
Slag	300	29	0.2
Fly ash	160	36	0.5
NaOH	3000	750	9.5
Na_2_SiO_3_	1300	671	9.4
CCR	80	0	0

**Table 10 materials-16-05360-t010:** The dosages of GBGBM and GBCBM with different binders (wt.% without water addition).

Mix Symbol	OPC	Slag	Fly Ash	NaOH	Na_2_SiO_3_	CCR
GBCBM	100	-	-	-	-	-
GBGBM	-	30	70	1.6	9.5	5

## Data Availability

Not applicable.

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
