# Peer review of "Preparation a High-Performance of Gangue-Based Geopolymer Backfill Material: Recipes Optimization Using the Taguchi Method"

_materials, 2023, doi:10.3390/ma16155360_

Round 1

Reviewer 1 Report

The paper discusses the use of gangue as the potential ingredients in the geopolymer as a backfill material. Despite the extensive works that had been conducted, the manuscript lacks of readability. It has many repeated abbreviations, the analysis mostly focuses at the increasing and decreasing trend in the data, and the research conduct has several critical flaws in it. I suggest that the authors take these comments into consideration when they decide to revise the manuscript.

1. Please send the manuscript for another professional proofreading process. Recheck for any unclear sentence, e.g. Line 143: Amorphous gel is the main component of the slag, and the main component is amorphous phase. The repeating information is not necessary, and if the slag was in powder form during XRD analysis, how the gel form was produced?

2. Line 201: How to measure slump at 3d and 28d ?

3. Line 238: How the mixing process was conducted by following Taguchi test procedure ? The subsequent procedure seems a normal concrete mixing procedure, what makes it different with the Taguchi test procedure ?

4. The critical error is by using a controlled curing process to produce the specimen. If the material is intended to be used as a backfilling material, how it will be cured on site ?

5. In the SEM sample preparation method, the method has serious flaw. If you used fragment from UCS specimen, how you can be sure that the microcracks that are discussed in the subsequent sections were preexist before UCS test and not due to the loading from UCS machine?

6. The word "rate" in sand rate and contribution rate seems not suitable to be used in this context.

7. Please add more in-depth discussion

The manuscript requires a comprehensive review, with a particular emphasis on enhancing the readibility and clarity. It is essential to consistently consider the reader's perspective throughout the writing process.  There are numerous unclear sentences, such as those found in Line 143, Line 201, Line 238, Line 245, and elsewhere. Please send the manuscript to a professional proofreading service.

Author Response

Dear reviewer:

       Many thanks for the comments and suggestions for the paper. The following are the answers and revisions I have made in response to the questions and suggestions. Words in blue are the changes I have made in the text.

Reviewer 2 Report

The submitted manuscript is about to find out the optimal combination of GBGBM by taking account the effects of grading, sand rate, w/b ratio, and a/b ratio on the slump, as well after 3 and 28 days UCS of GBGBM. The submitted research seems interesting and can be considered for publication but some points have remained ambiguous and must be discussed. I kindly ask authors to prepare a point-by-point rebuttal response letter and must be subjected to the manuscript as well, considering the following comments:

1)     Please make more comprehensive conclusion as in the revised version the following points must be included; materials and methods, the significant of this study, the scope of the effort, the procedures used to execute the work, and the major findings.

2)     In line 63-65 has been mentioned that the fly ash, slag, and other aluminosilicate materials can be processed using alkali…. So why metakaolinte has not been used as high reactive calcined clay in this study? If used the introduction must be more comprehensive and report about its structure and its reactivity? How about illite, muscovite, smectite and cholorite utilization? Following manuscripts can be helpful for metakaolinite reactivity and different forms of metakaolinite and their dehydroxylation temperature and its reactivity. https://doi.org/10.3390/ma16051863,https://doi.org/10.3390/nano13071196.

3)     What specific challenges are associated with the costly and weak filling materials in the strip filling method for underground reservoirs?

4)     How does the gangue-based geopolymer backfill material (GBGBM) aim to address the challenges mentioned?

5)     Can you explain the Taguchi method and its application in designing the series of experiments?

6)     What are the potential implications and benefits of achieving high-strength GBGBM for water storage in underground reservoirs?

7)     Besides the slump and unconfined compressive strength (UCS), are there any other characteristics or properties that were evaluated for GBGBM in the study?

8)     How were the optimal combination of GBGBM factors determined using the Signal-to-Noise ratio (S/N) based extreme difference and variance analysis?

9)     What are the specific hydration products and microstructural features that differentiate GBGBM from gangue-based cement backfill material (GBCBM)?

10)  Can you provide any insights into the mechanism or process through which GBGBM exhibits advantages such as reduced CO2 emissions, decreased energy consumption, and lower cost?

11)   Considering the sustainability and cost-effectiveness aspects, are there any limitations or potential drawbacks associated with the use of GBGBM?

12)  In practical applications, what other considerations or factors should be taken into account when using GBGBM for strip filling in underground reservoirs?

13)  Either use Fig. or Figure throughout the manuscript. 

Author Response

(The authors gave the same response as above.)

Reviewer 3 Report

Overall, the manuscript demonstrates good English language proficiency. The writing is clear and effectively conveys the research findings. However, there are a few instances where improvements can be made to address some repetitions within the text. These repetitions may be revised to enhance the flow and readability of the manuscript.

Author Response

(The authors gave the same response as above.)

Round 2

Reviewer 2 Report

The requested modifications have been implemented and the paper can be accepted for publication in the present form.